# Local and remote climate impacts of future African aerosol emissions

Christopher D. Wells[1,2], Matthew Kasoar[3], Nicolas Bellouin[4], Apostolos Voulgarakis[3,5]

[1]The Grantham Institute for Climate Change and the Environment, Imperial College London, London, UK
[2]School of Earth and Environment, University of Leeds, Leeds, UK
[3]Leverhulme Centre for Wildfires, Environment and Society, Department of Physics, Imperial College London, London, UK
[4]Department of Meteorology, University of Reading, Reading, UK,
[5]School of Environmental Engineering, Technical University of Crete, Chania, Greece

*Correspondence to*: Christopher D. Wells[1] (c.d.wells@leeds.ac.uk)

**Abstract.** The potential future trend in African aerosol emissions is uncertain, with a large range found in future scenarios used to drive climate projections. The future climate impact of these emissions is therefore uncertain. Using the Shared Socioeconomic Pathway (SSP) scenarios, transient future experiments were performed with the UK Earth System Model UKESM1, to investigate the effect of African emissions following the high emission SSP370 scenario as the rest of the world follows the more sustainable SSP119, relative to a global SSP119 control. This isolates the effect of Africa following a relatively more polluted future emissions pathway. Compared to SSP119, SSP370 projects higher non-biomass burning emissions aerosol emissions, but lower biomass burning emissions, over Africa. Increased SW absorption by black carbon aerosol leads to a global warming, but the reduction in the local incident surface radiation close to the emissions is larger, causing a local cooling effect. The local cooling persists even when including the higher African $CO_2$ emissions under SSP370 than SSP119. The global warming is significantly higher by 0.07 K when including the nonBB aerosol increases, and higher still (0.22 K) when including all aerosols and $CO_2$. Precipitation also exhibits complex changes. Northward shifts in the Inter-Tropical Convergence Zone (ITCZ) occur under relatively warmer northern hemisphere land, and local rainfall is enhanced due to mid-tropospheric instability from black carbon absorption. These results highlight the importance of future African aerosol emissions for regional and global climate, and the spatial complexity of this climate influence.

## 1 Introduction

Emissions of aerosols and their precursors have substantial and complex impacts on climate, both locally to their emission location and further afield (Liu et al., 2018; Thornhill et al., 2021). Their overall negative radiative forcing has dampened the warming effect of greenhouse gases historically, though the exact magnitude of this dampening forcing is uncertain (Bellouin et al., 2020; IPCC, 2021). The future of this forcing is also uncertain, and dependent on future emissions scenarios, with the recent Shared Socioeconomic Pathways (SSPs) projecting a broader possible set of future emissions than the prior Representative Concentration Pathways (RCPs) (Gidden et al., 2019). The SSPs are widely used as possible future pathways in Earth System Models, including in the recent 6[th] Assessment Report of the IPCC (IPCC, 2021).

Aerosols can either absorb incident solar radiation and therefore exert a positive radiative forcing (primarily black carbon (BC)) or scatter the radiation, causing a negative forcing (e.g. sulfate and organic carbon (OC)). The net negative present-day

aerosol forcing arises from the larger role of scattering than absorbing aerosol (IPCC, 2021). The combined top-of-atmosphere radiative effect of OC and BC, however, is uncertain (Grandey et al., 2018; Jiang et al., 2020; Thornhill et al., 2021). The

balance between negative OC and positive BC forcings depends on factors such as the underlying surface albedo, the optical properties of the aerosols, and the vertical distribution of the aerosol load (Hodnebrog et al., 2014; Mallet et al., 2020; O'Connor et al., 2021; Westervelt et al., 2020).

Due to their short lifetime of several days, aerosol impacts are strongly dependent on their emission location (Persad & Caldeira, 2018). Aerosols can have substantial impacts on circulation patterns, including on monsoon systems through both

local and remote effects (Li et al., 2018; Shawki et al., 2018; H. Wang et al., 2016; Z. Wang et al., 2017; Wilcox et al., 2020), though these changes may not significantly influence their impact on the global scale (Johnson et al., 2019). Shifts in the Inter-Tropical Convergence Zone (ITCZ) caused by interhemispheric energy imbalances under hemispherically asymmetric aerosol emissions may have contributed to historical changes in the West African Monsoon (WAM) (e.g. Lelieveld et al., 2019; Westervelt et al., 2018; Zanis et al., 2020); future Northern Hemisphere aerosol emissions reductions are projected to cause a

consequent continued increase in Sahel rainfall (e.g. Baker et al., 2015; Scannell et al., 2019; Shindell et al., 2012). The WAM is also sensitive to other factors such as Atlantic SST changes (Chadwick et al., 2017; Knippertz et al., 2015; Hill et al., 2017; Dong and Sutton, 2015). There is substantial disagreement in future changes of the WAM in models of the Coupled Model Intercomparison Project phase 6 (CMIP6) generation, particularly in the Western region (Chen et al., 2020; Almazroui et al., 2020). The local rainfall analyses in this study instead focus on areas to the south and east of the main WAM region.

Impacts of aerosol emissions on precipitation are particularly complex. Increases in scattering aerosols tend to decrease precipitation via their cooling effect, with this manifesting on multidecadal timescales; absorbing aerosols also tend to decrease precipitation, since their SW absorption represents a net source of atmospheric energy, reducing the energy deposited by latent heat (Liu et al., 2018; Richardson et al., 2018). At the global level, these effects are energetically constrained through TOA and surface fluxes, but on regional scales horizontal energy transport can influence this balance. Zhang et al., (2021) found

that higher African BC aerosol can increase precipitation locally, with the increased atmospheric energy from both BC absorption and this increase in latent heat balanced by horizontal energy transport.

The climate response to global and regional aerosol changes has been studied extensively in model experiments, using both idealised, instantaneous emissions perturbations, and transient, scenario projections. Idealised perturbations have been applied both globally and regionally across some key areas (Kasoar et al., 2018; Lewinschal et al., 2019; Samset et al., 2016; Yang et

al., 2019), but the influence of tropical regions remain understudied in this regard. The effects of various differing aerosol emissions scenarios on the climate have also been studied (Acosta Navarro et al., 2017; Allen et al., 2021; Tebaldi et al., 2021). However, the effects of single regions following different SSP emissions trajectories to other areas have not been studied using CMIP6, models and the effect of African emissions remains understudied.

The recent SSP scenarios used in CMIP6 project a wider range in future aerosol emissions than previous scenarios such as the

RCPs used in CMIP5 (Gidden et al., 2019), which may have been unrealistically narrow in their projections (Partanen et al.,

2018). This implies a wider range of possibilities for the impact of aerosols on the climate system than previously suggested. The climate impact of regional variations in these emissions scenarios has not yet been studied in detail.

The Tropics cover half the Earth's surface and are a substantial source of aerosol emissions, from a complex range of natural and anthropogenic sources. These emissions could therefore have a substantial effect on local and remote climates. The large population centres within the Tropics suggest that the societal impacts of any local climate and atmospheric composition changes would be substantial too. The human health impacts of particulate matter and ozone air pollution under the African aerosol scenario experiments studied here will be analysed in a separate paper.

This study applies time-varying SSP scenarios differing only in aerosol and reactive gas emissions over a single continent – Africa – in a fully-coupled Earth System Model, and investigates the impact on local and remote climates. Large, idealised aerosol emissions perturbations were also applied over Africa and the Tropics, and used to inform the analysis of the more complex transient experiments. The major results presented in the main text relate to the scenario experiments. These experiments are focused on due to the societal importance of realistic future emissions scenarios; the analysis has been substantially aided by the results of the idealised perturbations which are presented primarily in the Supplementary Material. The model used and experiments performed are described in Section 2, the temperature and precipitation responses and their mechanisms are detailed in Section 3, and discussion and conclusions are presented in Section 4.

## 2 Methods

This study uses the UK Earth System Model version 1, UKESM1, a participant in CMIP6 with a horizontal resolution of 1.875° x 1.25° and 85 vertical levels (Sellar et al., 2019). The model couples the ocean, atmosphere, and land components to Earth System components such as dynamic vegetation and ocean biogeochemistry. Aerosols are represented by the GLOMAP-mode 2-moment scheme, which simulates aerosol number and size in five lognormal modes (Mulcahy et al., 2018). The aerosol scheme is coupled to the interactive StratTrop chemistry scheme (Archibald et al., 2020). UKESM1 simulates black carbon (BC), organic carbon (OC), sulfate, sea salt, primary marine organic aerosol (PMOA), secondary organic aerosol (SOA), and dust aerosol, with dust simulated as an external mixture in 6 size bins using an earlier aerosol scheme, CLASSIC (Bellouin et al., 2011).

UKESM1 represents the Earth's climate well, with biases generally comparable to other CMIP6 models (Sellar et al., 2019). Its Equilibrium Climate Sensitivity (the steady-state warming realised upon a doubling of $CO_2$ concentrations) is 5.4K (Sellar et al., 2019). This is higher than most other CMIP6 models, with an average across models of 3.9K (Andrews et al., 2019; Zelinka et al., 2020), and higher than the AR6 very likely range of 2-5K. It exhibits the Double-ITCZ Bias (Sellar et al., 2019), common among similar models (Tian & Dong, 2020); its overall precipitation representation is reasonable, but like many models it struggles to accurately represent historically observed global temperatures. UKESM1's BC aerosol forcing is the strongest of all models participating in AerChemMIP, but a strong negative sulfate forcing gives an overall aerosol forcing of

-1.09 ± 0.04 Wm$^{-2}$ (O'Connor et al., 2021), close to the multimodel value of -1.01 ± 0.25 Wm$^{-2}$ (Thornhill et al., 2021). The simulated AOD is low over West Africa when compared to satellite and ground observations (J. Mulcahy et al., 2020), a common bias in climate models (Wilcox et al., 2020).

The Shared Socioeconomic Pathway (SSP) emissions scenarios, created for use in CMIP6, are used in this study to generate transient 21st century emissions pathways. They comprise a set of future spatially-resolved emissions trajectories, given particular socioeconomic and climate change mitigation trends. They are denoted SSPx-y, with x an integer from 1 to 5 indicating a different socioeconomic baseline (O'Neill et al., 2017), and y indicating the approximate top-of-atmosphere radiative forcing in 2100 (van Vuuren et al., 2014). The two scenarios used in this experiment are SSP119 and SSP370. The first of these corresponds to strong climate mitigation, following the "Sustainability" socioeconomic trends in SSP1 and a forcing of 1.9 W/m2 in 2100, approximately compatible with the more ambitious Paris Agreement goals (O'Neill et al., 2016). The second scenario instead assumes the "Regional Rivalry" associated with SSP3, coupled with weak mitigation of both greenhouse gases and aerosols, leading to a radiative forcing of 7.0 w/m2 in 2100 and high aerosol levels (Gidden et al., 2019). In this study, the control scenario experiment consists of the whole globe (including Africa) following the high-mitigation SSP119 scenario. Additional experiments are carried out using hybrid scenarios of SSP119 and SSP370. In all of them, the world outside of Africa follows SSP119 in its entirety. Africa, however, follows the SSP370 pathway for different subsets of emissions. Five experiments are performed, named after the set of emissions for which Africa follows SSP370. The aerosol species altered are sulfate (via emissions of $SO_2$), OC, and BC. The reactive gases are Carbon monoxide (CO), Nitrogen oxide (NO), Ammonia ($NH_3$), Ethane ($C_2H_6$), Propane ($C_3H_8$), Di-methyl sulfide, Formaldehyde (HCHO), Acetone ($C_3H_6O$), Acetaldehyde ($C_2H_4O$), and non-methane volatile organic compounds (NVOCs); these species are altered, and the list provided, for completeness, but the aerosol impacts drive the response under these experiments. All the species have biomass and non-biomass components in UKESM1 except $SO_2$, which has no biomass component.

| | BB OC+BC and reactive gases | nonBB OC+BC and reactive gases | $SO_2$ | $CO_2$ |
|---|---|---|---|---|
| Control | **SSP119** | **SSP119** | **SSP119** | **SSP119** |
| AerAll | *SSP370* | *SSP370* | *SSP370* | **SSP119** |
| AerBB | *SSP370* | **SSP119** | **SSP119** | **SSP119** |
| AerNonBB | **SSP119** | *SSP370* | *SSP370* | **SSP119** |
| $CO_2$ | **SSP119** | **SSP119** | **SSP119** | *SSP370* |
| AerAllCO$_2$ | *SSP370* | *SSP370* | *SSP370* | *SSP370* |

Table 1: The scenario followed for African emissions of aerosols and reactive gases, and concentrations of $CO_2$, in each of the scenario experiments. Non-African emissions and concentrations of all species are taken from SSP119 in every experiment. BB = biomass burning; OC = organic carbon; BC = black carbon.

The scenario experiments are named with the species for which SSP370 is used for African emissions or concentrations. Table 1 indicates the experiments, and the scenario used for the emissions and concentrations of each species. "'AerAll' refers to the experiment where all aerosols and reactive gases follow the SSP370 scenario over Africa. "AerBB" refers to Biomass Burning aerosols and reactive gases; "AerNonBB" is then those emissions from non-Biomass Burning sources, i.e. fossil fuels and

biofuel use. In the "$CO_2$" experiment, globally averaged $CO_2$ concentrations were increased from the SSP119 concentrations in the control, to approximately the levels reached globally if Africa had followed SSP370 emissions of $CO_2$ whilst the rest of the world followed SSP119. These $CO_2$ concentrations were calculated using the MAGICC6 simplified climate model (Meinshausen et al., 2011). Finally, "AerAllCO$_2$" perturbs all aerosols, reactive gases, and $CO_2$ concentrations such that Africa follows SSP370. Each experiment was carried out from 2015-2100, and multiple ensemble members of each were used, each

with slightly different initial atmospheric and ocean conditions to explore the internal climate variability. These ensembles were: ten members of the SSP119 control (five newly simulated for use here, and five used from the pre-existing CMIP6 ScenarioMIP experiments (Tebaldi et al., 2021)), and seven members of each of the five additional experiments.

While the transient scenario experiments are the focus of this study, idealised aerosol emissions perturbation experiments were also performed with UKESM1. These involved changes in similar species over similar regions as in the scenario experiments,

but were driven by much bigger emissions perturbations, which were sustained constantly until the climate had approximately stabilised. This allows for a clearer understanding of the general climate response to the emissions changed in the scenario experiments. As these idealised experiments had to be computed from an equilibrium climate, UKESM1 was initialised from 2015 and ran with present-day (2006-2015 average) emissions of aerosols and reactive gases and present-day (2015) concentrations of greenhouse gases. The model was judged to have approximately reached equilibrium in global temperatures

after 135 years (model year 2150), and the idealised emissions perturbations were then applied:

10×Trop-SO$_2$ (SO$_2$ emissions multiplied by 10 in the Tropics, defined as 30° S to 30° N)

zeroTropBB-OCBC (Biomass Burning (BB) OC and BC emissions removed in the Tropics)

10×TropBB-OCBC (BB OC and BC emissions multiplied by 10 in the Tropics)

10×AfricaBB-OCBC (BB OC and BC emissions multiplied by 10 over Africa)

These perturbations were applied for 200 years; the first 50 years were discarded as spin-up, as the climate system adjusts to the applied forcing, and the final 150 years are analysed as the equilibrium response. This method is consistent with prior

studies which used similar spin-up times to reach equilibrium under similar-sized aerosol perturbations in earlier generation

models (Kasoar et al., 2018; Myhre et al., 2017; Shawki et al., 2018). Six control members, five of zeroTropBB-OCBC, and two each of the experiments with large emission increases were simulated. To more effectively isolate the climate response to the aerosol changes, in these idealised experiments the vegetation was held at present-day levels, and the link between changing CH4 concentrations and CH4's radiative forcing was also deactivated. Thus, no climate impacts of changes in vegetation or

CH4 compared to the present-day were simulated in these idealised experiments. In contrast, the scenario experiments used the standard UKESM1 CMIP6 configuration, with the radiation and vegetation schemes unaltered.

Figure 1 shows the emissions changes applied in the scenario experiments. Total OC+BC emissions are higher in the AerAll experiment than in the control (panel e), while the BB aerosol subset (c) shows the opposite trend. This is because the nonBB emissions (a) – from fossil fuel and biofuel burning – are straightforwardly related to the SSP scenario, with higher emissions

under weaker climate mitigation, whereas the link between BB emissions and the scenario drivers is more complex. The use of different Integrated Assessment Models to generate the different scenario emissions (IMAGE for SSP1 (van Vuuren et al., 2017) and AIM/GCE for SSP3 (Fujimori et al., 2017)) precludes a complete understanding of the exact causes of the relatively higher African BB emissions projected in SSP119 than SSP370, even when consulting the model teams, but the dominant component of the total emissions change in AerAll is the higher nonBB emissions in SSP370. NonBB carbonaceous aerosol

emissions (g) are concentrated near highly populated and industrial areas, primarily in West and East Africa as well as the North coast. BB emissions are centred in the areas of seasonal BB activity north of the Equator in DJF and south of it in JJA. The difference in total African $SO_2$ emissions between the scenarios is small, with relatively higher emissions in northern Africa in SSP370 offset by projected higher industrial $SO_2$ emissions in SSP119 in the south of the continent (except South Africa). Global $SO_2$ emissions (b) drop quickly overall in both experiments, with only a small magnitude difference by 2100.

The total aerosol emissions difference between the scenarios applied in this study is therefore dominated by changes in carbonaceous aerosol. The $CO_2$ concentration (f) increases in both scenarios from the present day, before peaking around mid-century. It drops below 400ppm in the (SSP119) control, but when switching Africa's emissions from SSP119 to SSP370 the concentration peaks later and decreases less sharply, remaining above 2015 levels in 2100.

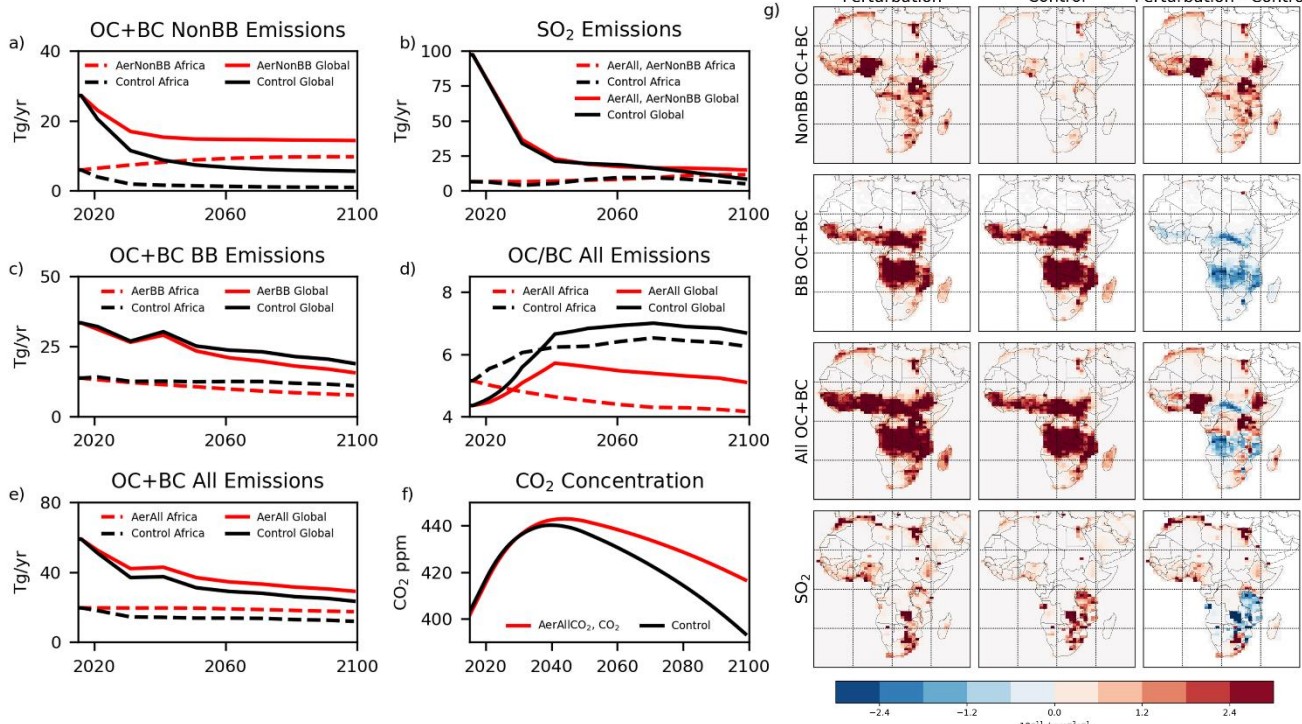

Figure 1: Timeseries of OC+BC (carbonaceous) aerosol, from nonBB (a), BB (c) and their sum (e), as well as $SO_2$ emissions (b), and the OC/BC ratio for total emissions (d). These emissions timeseries are shown globally and over Africa for OC+BC. $CO_2$ global concentrations are also indicated (f). The control experiment follows the black curves, with the changed emissions labelled according to the experiment(s) they relate to. Also shown are maps of aerosol emissions in each scenario, and their difference, in 2060 (g).

## 3 Results

This section details the climate response to the different emissions changes, focussing on the scenarios, particularly for the period from 2070-2100. This period is selected to maximise the difference between the climates, as the emissions diverge throughout the century, while reducing internal variability through averaging over a 30-year period as well as across ensemble members.

Figure 2 shows the global mean surface temperature in each experiment, from both the idealised and scenario experiments. Global and regional temperatures in 2070-2100 relative to a pre-industrial control UKESM1 experiment are also indicated in

Table 2 for the scenario experiments. Beginning with the idealised experiments (panel a), as detailed in the methodology, UKESM1 was forced with present-day emissions for 135 years to reach equilibrium, before applying the idealised perturbations for 200 years. The increased sulfate experiment, 10×Trop-SO$_2$, resulted in a strong global cooling of −1.93 ± 0.07 K relative to the control, consistent with the strong negative forcing associated with sulfate aerosol (Smith et al., 2020) and with the size of the perturbation applied here. When carbonaceous BB aerosol is increased in 10×TropBB-OCBC and

10×AfricaBB-OCBC, however, a substantial global warming of 1.08 ±0.07 K and 0.78 ± 0.07 K relative to the control occurs, respectively, while the weaker and opposite zeroTropBB-OCBC experiment causes a slight global cooling, with a temperature change of −0.07 ± 0.05 K. This is consistent with UKESM1's positive BC forcing being larger in magnitude than its negative OC forcing (O'Connor et al., 2021), and indicates that this holds even for BB emissions, despite the fact that they have a higher OC/BC ratio than other carbonaceous aerosol sources (von Schneidemesser et al., 2015).

The global mean temperature in the SSP119 control experiment (panel b) continues to increase from its historical trajectory, before levelling out around 2060 with a 2070-2100 temperature 2.27 K higher than pre-industrial times (Table 2). The experiment with increased CO$_2$ warms further, as expected, with a significantly higher warming of 2.43 K in 2070-2100 compared to the control (Table 2). AerNonBB also exhibits a significantly higher warming than SSP119, though the dampening effect of reduced BB emissions causes AerAll to not see a significant difference in the global temperature change. The largest

warming occurs in AerAllCO$_2$, where both the net warming aerosol and CO$_2$ are increased, giving a total global warming of 2.49 K, a level 0.22 K higher than the control.

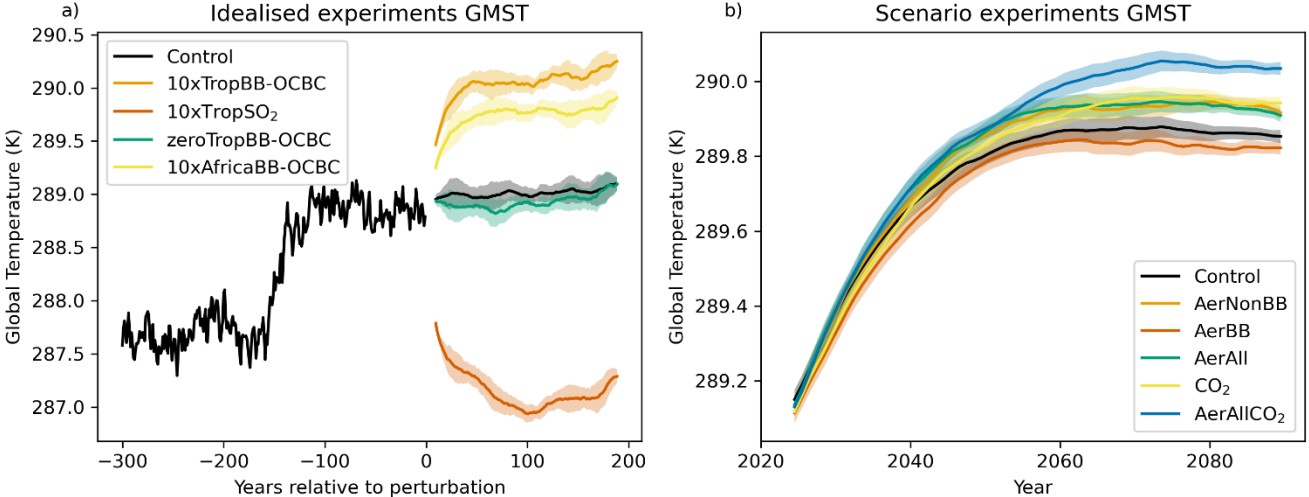

Figure 2: Global mean temperature response historically and under the idealised emissions perturbations (left) and through the 21st century in the emissions scenario experiments (right). 20-yr rolling means are shown, with the control inter-member standard deviation shown by the shaded envelopes.

| | Global | Africa | NTropAf | Arctic |
|---|---|---|---|---|
| Control | 2.27 | 2.50 | 2.41 | 6.68 |
| AerAll | 2.33 | *2.35* | *1.83* | **6.89** |
| AerBB | 2.28 | 2.51 | 2.43 | 6.76 |
| AerNonBB | **2.34** | *2.36* | *1.78* | 6.84 |
| $CO_2$ | **2.43** | **2.68** | **2.58** | **7.01** |
| AerAllCO$_2$ | **2.49** | 2.53 | *2.03* | **7.07** |

Table 2: Changes in surface temperature from pre-industrial times (calculated using a pre-existing PiControl simulation) to the period 2070-2100 in the control scenario and each experiment, globally and in several regions. Africa is defined as per the emissions region, visible in the difference panels in Figure 1; NTropAf (northern tropical Africa) is the high emissions region from 0° E-30° E and 0° N-30° N; the Arctic is 60° N-90° N. The values are bolded/italicised to indicate statistically significant higher/lower temperatures than the control experiment.

The effective radiative forcings (ERFs) of the idealised aerosol perturbations, calculated via fixed SST and sea ice experiments (see the Supplementary Material) reflect the temperature responses, with a positive ERF under increased carbonaceous aerosol and a negative forcing when increasing sulfate aerosol. ERFs were not calculated for the scenario experiments, as this would have required parallel fixed-SST and sea ice experiments. The TOA downward radiative fluxes can be calculated from the coupled runs, and are consistent with the impacts on temperature. For example, compared to the control, AerNonBB exhibits a clear-sky LW flux smaller by $-0.16 \pm 0.08$ W m$^{-2}$, due to its relatively warmer temperature. This is partially offset by a positive clear-sky SW forcing of $0.08 \pm 0.04$ W m$^{-2}$, concentrated over Africa and due to the increased BC absorption. AerAllCO$_2$ drives a more negative clear-sky LW flux of $-0.25 \pm 0.08$ W m$^{-2}$, due to its even warmer temperature under additional $CO_2$, with a similar clear-sky forcing to AerNonBB driven by the nonBB aerosols.

The effect on net surface shortwave radiation of the scenario experiments – except the experiment which only perturbed $CO_2$ – is displayed in Figure 3, for 2070-2100. While the AerNonBB experiment caused a global warming, the increased atmospheric aerosol burden reduced overall shortwave into the surface over and around Africa. A consistent and opposing increase in surface shortwave radiation occurs in AerBB due to the relative reduction in BB emissions, with AerAll reflecting the combination of these effects. The changes under AerAllCO$_2$ are consistent with AerAll, with aerosols dominating the response, with an additional positive surface forcing over the Arctic due to the reduced albedo under sea ice melt.

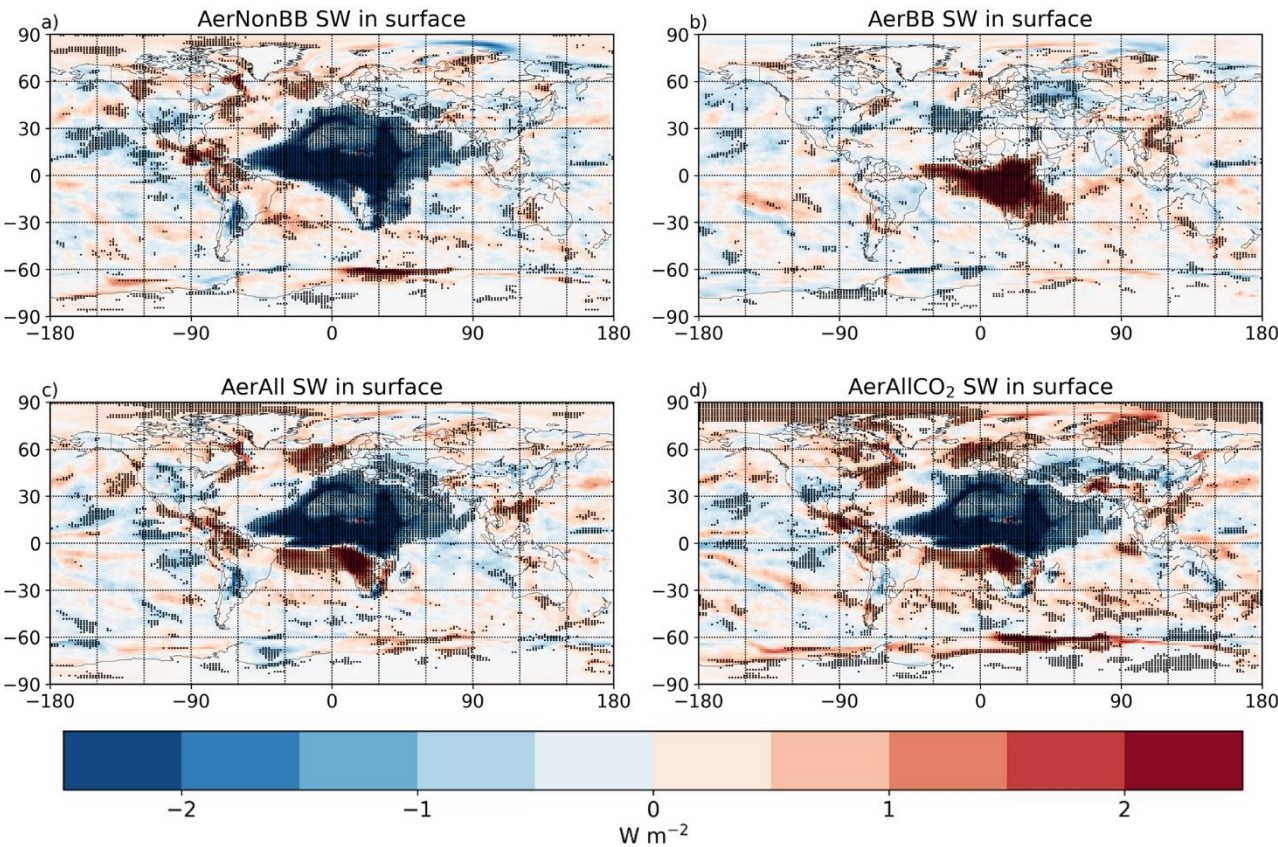

Figure 3: Effect of each scenario experiment (except that changing only $CO_2$) on net downwards surface shortwave radiation in 2070-2100 relative to the control scenario. Stippling indicates gridcells where the change is significant with respect to intra-ensemble variation.

Turning to the spatial temperature response, Figure 4 shows maps of the 2070-2100 mean temperature response in DJF and JJA under each scenario experiment. The increased global $CO_2$ concentrations in the $CO_2$ experiment lead to a general warming (panels g, h). While the increased aerosol experiments (a, b, e, f, i, j) drive a general warming effect, a striking local cooling over much of Africa is found. This is consistent with the reduced net downwards surface shortwave radiation. The cooling

occurs because absorbing aerosol, while increasing the energy within the Earth system and therefore raising temperatures globally, acts to reduce the radiation into the surface underneath it (adding to the cooling effect of OC), which here outweighs the general warming effect close to the perturbation, where the burden change is largest. This local surface cooling even outweighs the enhanced warming from additionally increased $CO_2$ in AerAllCO2. Whereas the local cooling from increased nonBB aerosol occurs around the main nonBB emissions areas in northern tropical Africa in both seasons, the AerBB

experiment (c, d) exhibits the BB emissions seasonal cycle in its local warming. This warming occurs since BB aerosol emissions are lower under this experiment, hence increasing the incident surface radiation. It should be noted that the local

cooling pattern does not match exactly the pattern of emissions changes between the scenarios; the reason for this is linked to the background circulation pattern, and is explored later in this paper. An analogous figure showing the surface temperature response to the idealised perturbations is given in Supplementary Figure S1. Enhanced local warming is present in

southwestern Africa under AerAll; this is due to the local reduction in $SO_2$ emissions (Figure 1), increasing the incident surface radiation. Statistically significant remote impacts – warming/cooling under increased/decreased carbonaceous aerosol – occur across broad land and ocean regions, more so under the larger nonBB-changing experiments, and with almost all remote areas experiencing warming under AerAllCO2.

Table 2 shows the temperature change from pre-industrial times to the period 2070-2100 over several regions in addition to

the global mean, for the control scenario and each experiment. The Arctic experiences substantially more warming than the global mean, occurring predominantly in DJF (Figure 4) as is typical of Arctic amplification (Stjern et al., 2019). Africa, defined using the emissions region as can be seen in Figure 1, experiences larger overall warming since pre-industrial times than the global mean, consistent with the larger land than ocean response. However, while the $CO_2$ experiment exhibits enhanced African warming relative to the control, the increased nonBB aerosol in AerNonBB and AerAll drives less overall

warming over Africa than in the control scenario, by around 0.15K. Focussing further, on the strong cooling region from 0° E-30° E, 0° N-10° N, this reduction in warming reaches 0.6K in AerNonBB, reducing the overall warming trend from pre-industrial times to 2070-2100 by 25%.

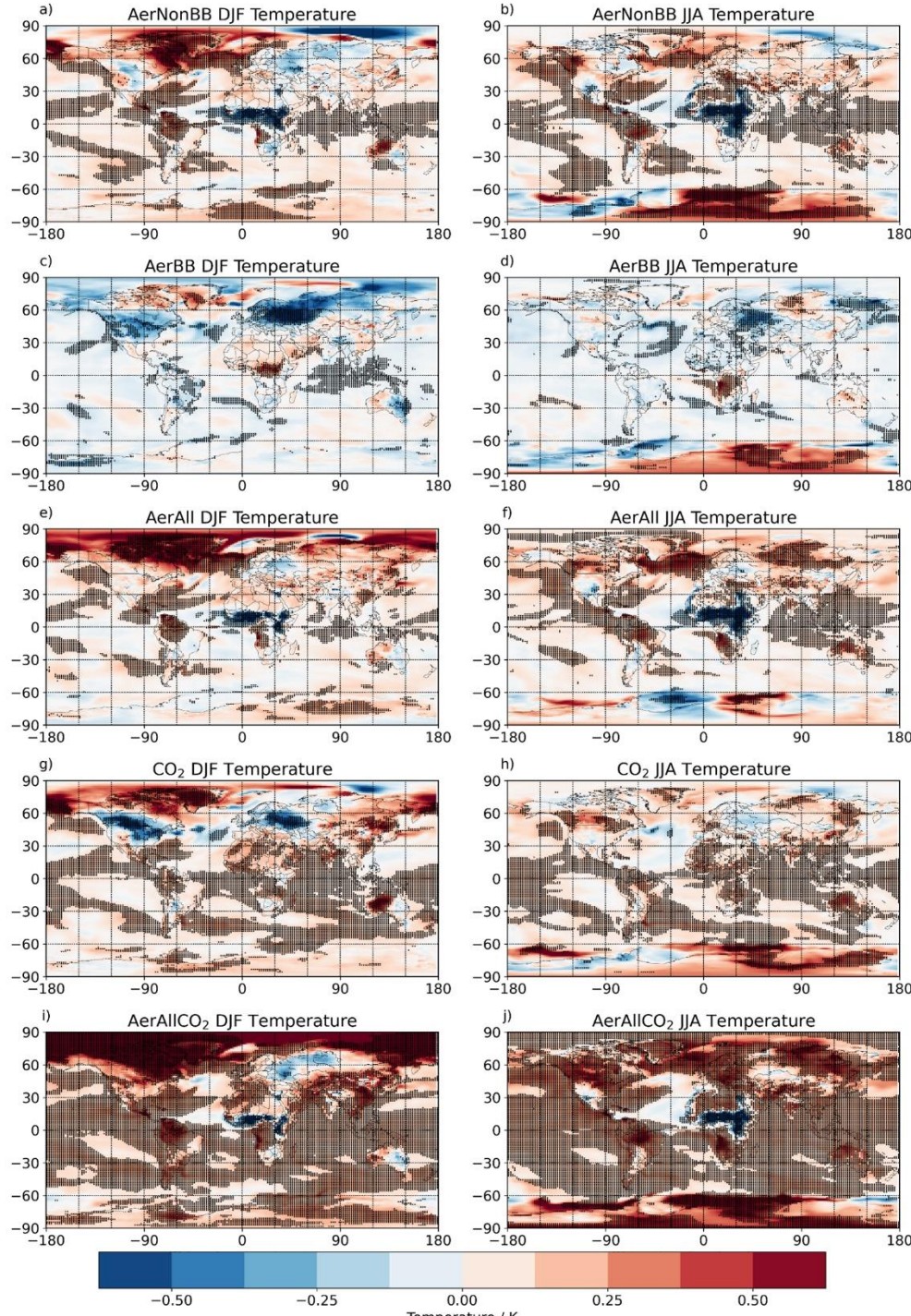

Figure 4: Effect of each scenario experiment on surface temperatures in DJF (left) and JJA (right) in 2070-2100. Stippling indicates gridcells where the change is significant with respect to intra-ensemble variation.

Turning to the effects of the scenario experiments on atmospheric circulation patterns, Figure 5 shows the 2070-2100 response of vertical velocity at 850hPa under each of the aerosol-only experiments in DJF and JJA, as well as the background vertical velocity in the control simulations. Figure 6 shows the Atlantic zonal-mean response and background. The ITCZ can be clearly distinguished in DJF at 0° N-10° N in the control over Africa and the Pacific and slightly further south over the Atlantic (Figure 5g), and north of the DJF position, particularly over land, in JJA (panel h). Since the overall effect of the carbonaceous aerosol in the experiments is a positive forcing (to generate the overall warming in Figure 2b), and the increase in nonBB aerosol is primarily north of the Equator (see Figure 1), the experiments with increased nonBB represent an overall positive northern hemispheric forcing. Positive radiative forcings act to pull the ITCZ location towards the hemisphere where they occur (Schneider et al., 2014; Voigt et al., 2017; W Frierson & Hwang, 2011), so a northward shift in the ITCZ can be expected under the increased African nonBB aerosol experiments here in DJF, though the effect in JJA may be complicated by the more northern background ITCZ location. This is indeed clear in DJF in the Pacific, Atlantic, and Africa in the 850hPa vertical velocity maps (Figure 5 a and e), and also in the Atlantic zonal-mean panels in AerAll and AerNonBB (Figure 6 a and e). In the less drastic AerBB experiment, the BB aerosol emissions decrease in DJF is also centred north of the Equator, suggesting a southward ITCZ shift due to overall cooling in the northern hemisphere in this case. This southward shift is present over the Atlantic zonally (Figure 6 panel a), and over Africa in panel c of Figure 5, but less clear over the Pacific, likely due to the smaller emissions difference under this experiment. In JJA, slight northward shifts are present in eastern Africa and the Atlantic (Figure 5d; Figure 6b). Figure S2 in the Supplementary Material shows consistent, stronger effects under the much larger idealised emissions perturbations.

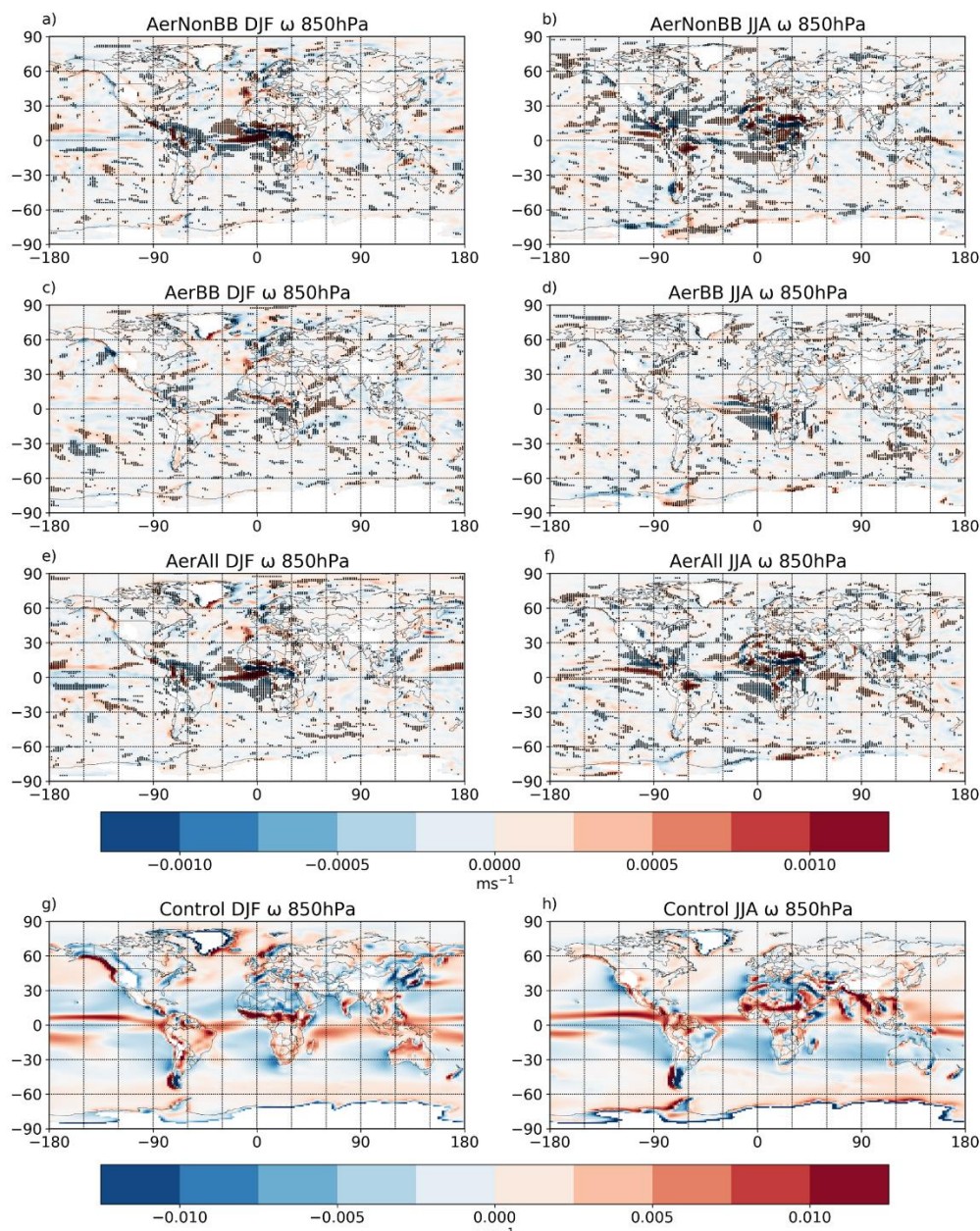

Figure 5: Vertical velocity changes at 850hPa in DJF and JJA, for the three aerosol perturbations (AerBB, AerNonBB, and AerAll) relative to the control, for the period 2070-2100. The control background is also shown (panels g and h). Stippling on panels a-f indicates gridcells where the change is significant with respect to intra-ensemble variation.

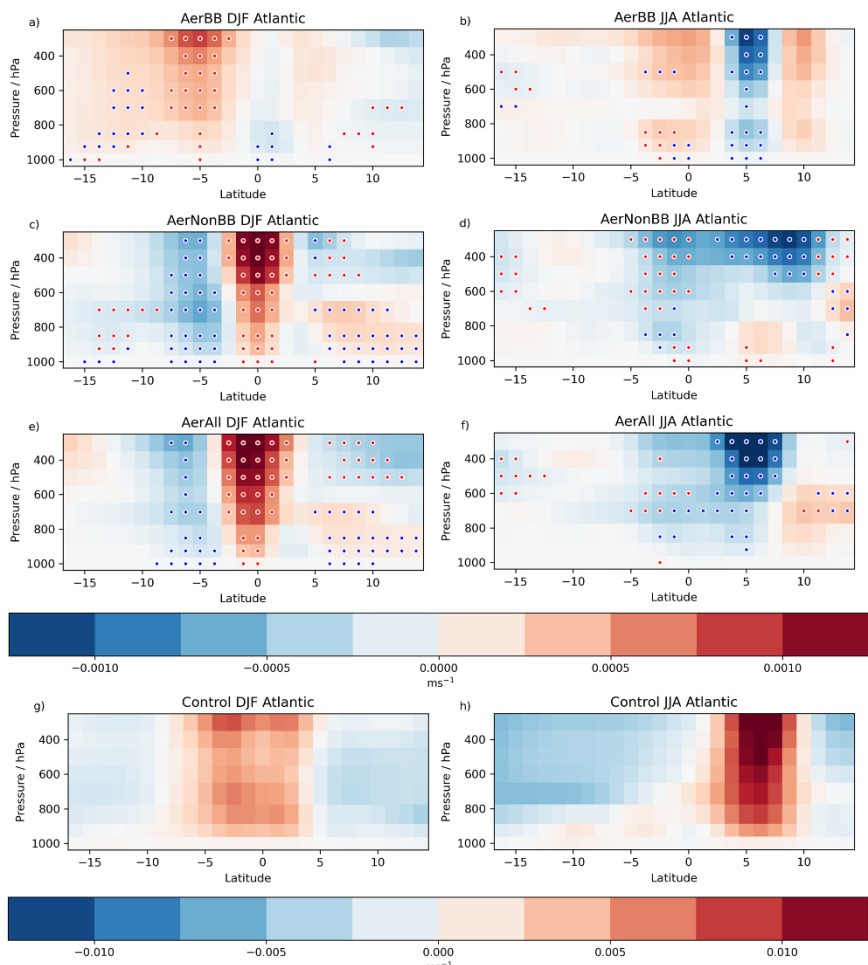

Figure 6: Vertical velocity changes zonally over the Atlantic in DJF and JJA, for the three aerosol perturbations minus the control (AerBB, AerNonBB, and AerAll), for the period 2070-2100. The control background is also indicated (panels g and h). The Atlantic here is 40° W − 15° W; narrower than the length at the Equator to avoid including land between 15° S and 15° N. Stippling indicates gridcells where the change is significant with respect to intra-ensemble variation, and is coloured

red when the change strengthens the background circulation, and blue when opposing.

The precipitation response in both DJF and JJA is then shown in Figure 7 for the aerosol experiments, with maps as well as global and Pacific zonal means, along with the control rainfall. Figure S3 in the Supplementary Material presents the same for the idealised experiments, with consistent results. The Pacific zonal mean is shown as the ITCZ is coherent here and extends

across a large longitude range, allowing any shifts to be more readily identified. Over the Pacific, the northward shift in DJF in the AerAll and AerNonBB experiments identified in the vertical velocity panels is also clear in the rainfall response, with a characteristic wetting/drying in the north/south of the control ITCZ band. No clear Pacific ITCZ shift manifests in JJA under

AerAll or AerNonBB, indicating that the more northern position of the ITCZ in JJA may prevent the perturbations from inducing a strong forcing asymmetry across the ITCZ as is the case for DJF. AerBB shows little clear Pacific precipitation
response, consistent with the vertical velocity change, but some local effects are found, discussed shortly. Northern South America experiences substantial decreases in precipitation under increased African nonBB aerosol (in AerAll and AerNonBB), centred over areas with large background rainfall in both DJF and JJA. While small in percentage terms, these are relatively large magnitude shifts, suggesting that through their impacts on the ITCZ location, aerosol emissions can produce substantial changes in precipitation in key locations remote from their emission. The representation of the West African Monsoon (WAM)
in UKESM1 features an anomalous second rainfall peak off the West African coast (Figure 7x), common among climate models (Raj et al., 2019). While increased WAM precipitation occurs under AerBB, consistent with a localised northward ITCZ shift (Figure 7f), the precipitation responses under changed nonBB emissions in AerNonBB and AerAll (Figure 7l, 7r) appear to be dominated by local effects. This differs from prior studies discussed in the introduction, which primarily applied perturbations to remote (i.e. non-African) aerosols, resulting in often substantial WAM shifts. This difference is due to the
relatively weak ITCZ shifts found here – since the perturbation straddles the equator – coupled with the strong local effects as emissions were perturbed within and near to the WAM region.

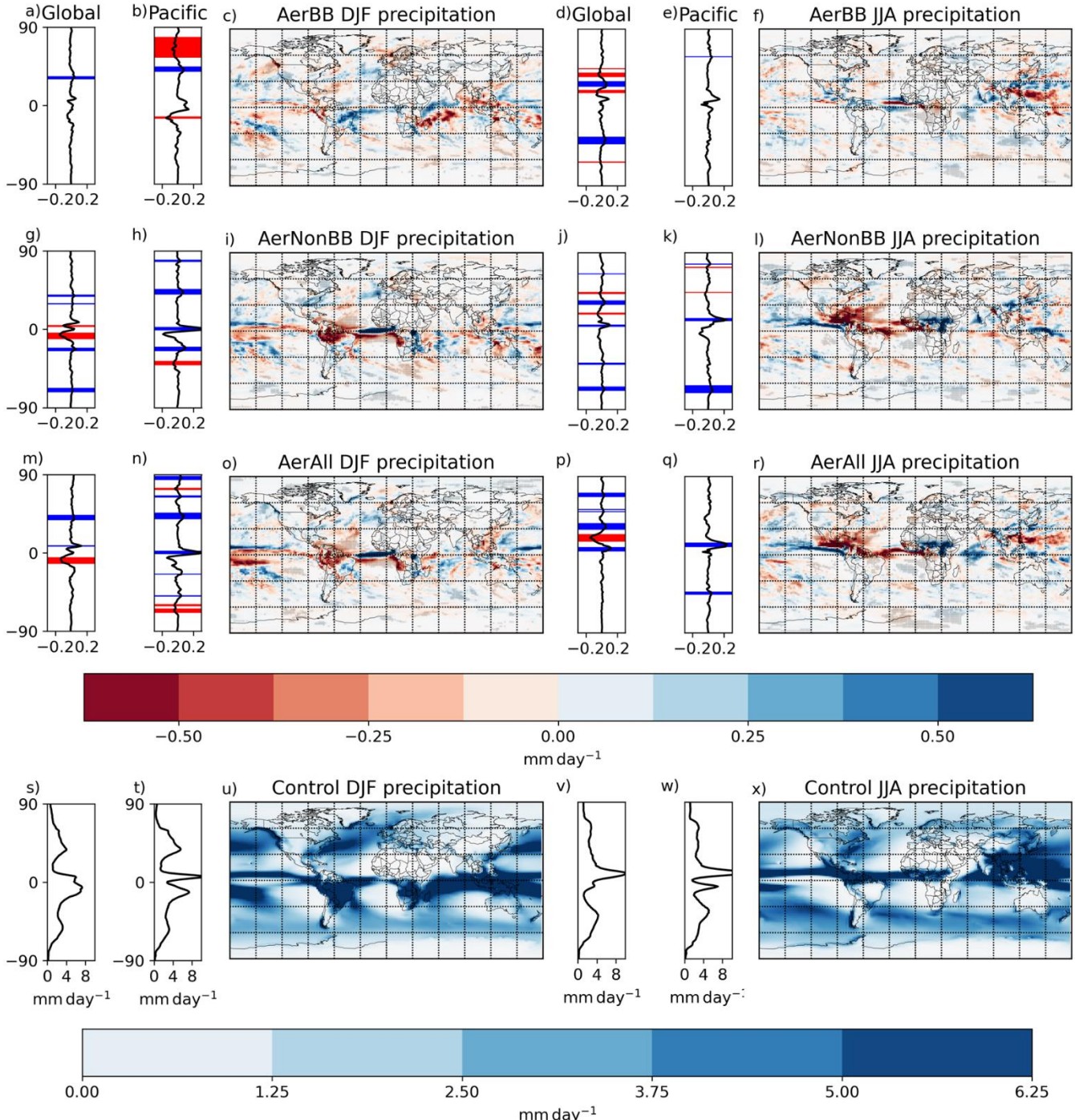

Figure 7: 2070-2100 precipitation response in AerAll (top), AerBB (middle), and AerNonBB (bottom) for DJF (left) and JJA (right), with maps and zonally both globally and across the Pacific (150° E − 270° E). Control DJF and JJA precipitation maps

are shown at the bottom. Stippling on the maps indicates gridcells where the change is significant with respect to intra-ensemble variation. The coloured areas in the zonal change panels show significance – red for decreased, blue for increased precipitation.

The local pattern of precipitation change over Africa is complex, with large increases under increased nonBB aerosol, in different regions in DJF and JJA. These responses can be understood through an investigation of the effect of the emissions changes on the vertical atmospheric structure.

Figure 8 shows vertical profiles of the change in carbonaceous aerosol burden, SW heating, potential temperature, atmospheric
stability, vertical velocity, and cloud, in 2070-2100 in AerNonBB. These profiles are given for three region-season combinations, with the regions indicated in the bottom right panel of Figure 9: Region 1 centres on the Eastern Sahel, with profiles shown for DJF and JJA, while Region 2 is centred on the high-emissions southern Nigeria-western Cameroon area and the Gulf of Guinea, with just the DJF profile shown. Figure 9 shows the background (control) 850hPa vertical velocity, and the AerNonBB-induced 2070-2100 change in three variables: temperature at 420m, vertical velocity at 6000m, and surface
precipitation, centred over Africa for DJF and JJA.
In Region 1 in JJA, the strong increase in emissions (Figure 1) coincides with a strong background upwards vertical velocity (Figure 9a). This allows the carbonaceous aerosol to increase strongly throughout the column (Figure 8b). This large aerosol burden then generates a strong SW heating vertical response (c), and also gives rise to the cooling at the surface and in the lower atmosphere, as seen in Figure 4 and Figure 9. Thus, the pattern of local cooling depends not just on the pattern of
emissions change, but also on the background circulation, with areas of strong uplifting circulation allowing for higher aerosol loads, and consequent reductions in radiation into the surface. The vertical profile of potential temperature change over Region 1 in JJA (d) reflects this: a significant cooling occurs up to 1.5km, giving way to warming at higher levels. Crucially, the combination of the receding cooling effect with height and increased SW absorption gives rise to a warming peak around 4km, with less warming above for several kilometres. The effect is therefore an instability anomaly in the mid-troposphere (e), which
gives rise to enhanced vertical velocity above 4km (f). This enhanced mid-tropospheric convection is strongly correlated spatially with the local cooling (Figure 9b). Cloud is enhanced above this (g), with reduced mid-level cloud coinciding with the SW heating. Precipitation is therefore strongly enhanced in these areas with increased mid-level convection, i.e. across northern equatorial Africa in JJA (Figure 9d), enhancing the background rainfall (Figure 7).
To summarise: Region 1 in JJA is associated with both increased carbonaceous aerosol emissions and background convection
exhibiting strong uplifting of the aerosol; SW heating from the BC aerosol leads to a warming peak, which generates a mid-tropospheric instability anomaly, and hence increased precipitation.
This mechanism can be applied to different region-season combinations. Figure 8 shows vertical profiles for Region 2 in DJF, which also features strong increased aerosol emissions and background convection. Thus, a warming peak occurs around 3km (panel j), convection is strongly enhanced above this (i), and precipitation is increased (Figure 9h). The horizontal circulation
also plays a role here, with the aerosol carried downwind generating a strong precipitation response into the Atlantic (Figure

9h). In contrast to the two region-season combinations initially analysed, Region 1 in DJF – while having strong emissions still – is dominated by descending air (Figure 8r). This strongly suppresses the vertical extent of the aerosol increase (n), precluding the mechanism described above from manifesting, and thereby preventing any significant precipitation response (Figure 9h). A further case study – over East Africa in DJF under the idealised emissions perturbations – is presented in Supplementary Figure S4.

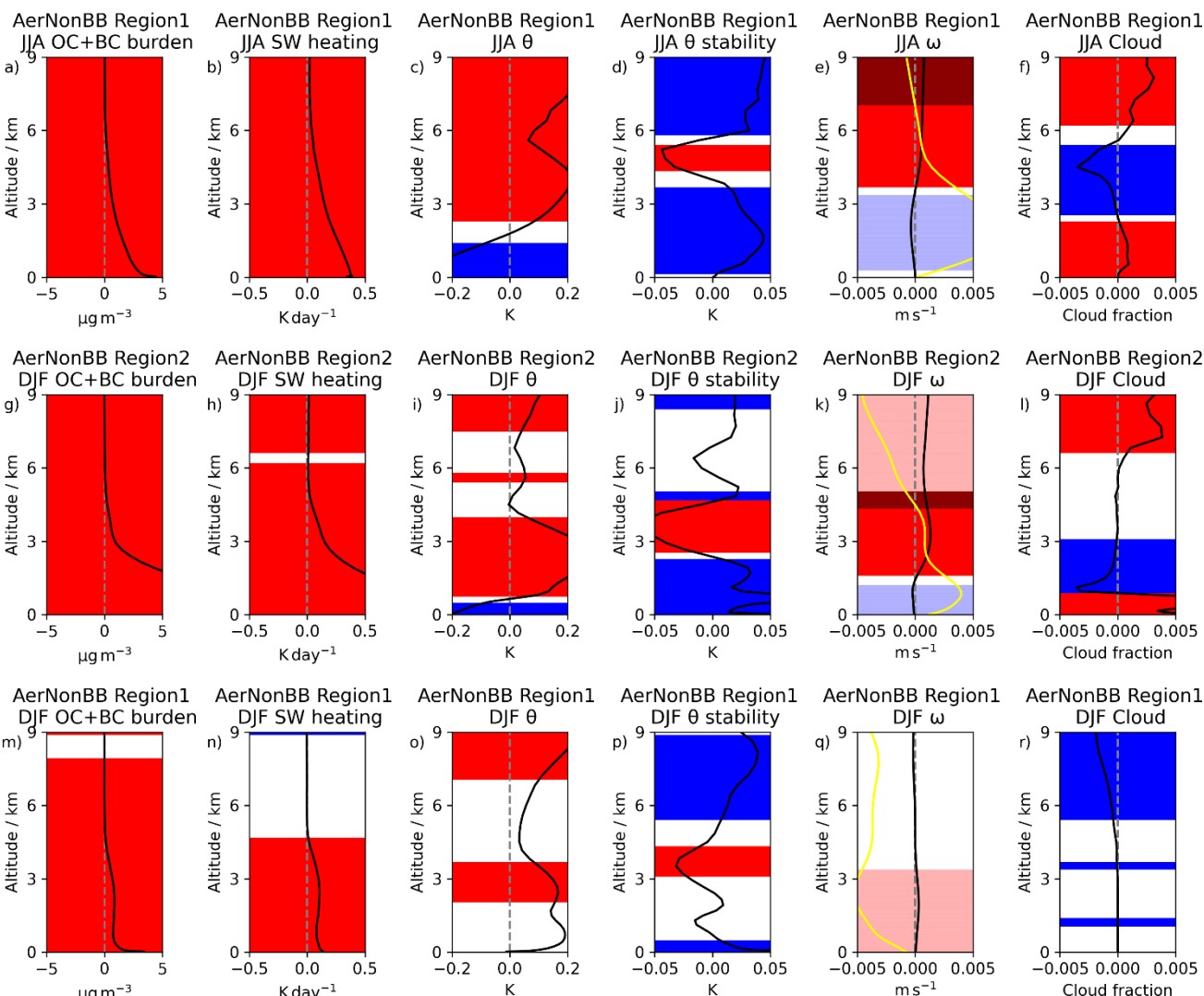

Figure 8: Changes in the vertical profile of carbonaceous aerosol burden (left column), clear-sky SW heating (2nd column), potential temperature (θ, 3rd column), potential temperature vertical gradient calculated at each level as the potential temperature at the height immediately above the level minus at that level (4th column), air vertical velocity (ω, 5th column), and cloud (6th column), over Region 2 in JJA (top), and Region 1 and Region 2 in DJF (middle and bottom), in 2070-2100 under the AerNonBB experiment relative to the control. The control vertical velocity profile is indicated in yellow. Coloured

horizontal lines indicate significant increases (red) or decreases (blue). The vertical velocity colour shades are modified depending on the sign of the change relative to the control baseline: lighter red and blue when the change is enhancing the background, normal when opposite in direction to the background but with the change weaker in magnitude, and darker when opposed with higher magnitude than the control, acting to reverse the sign.

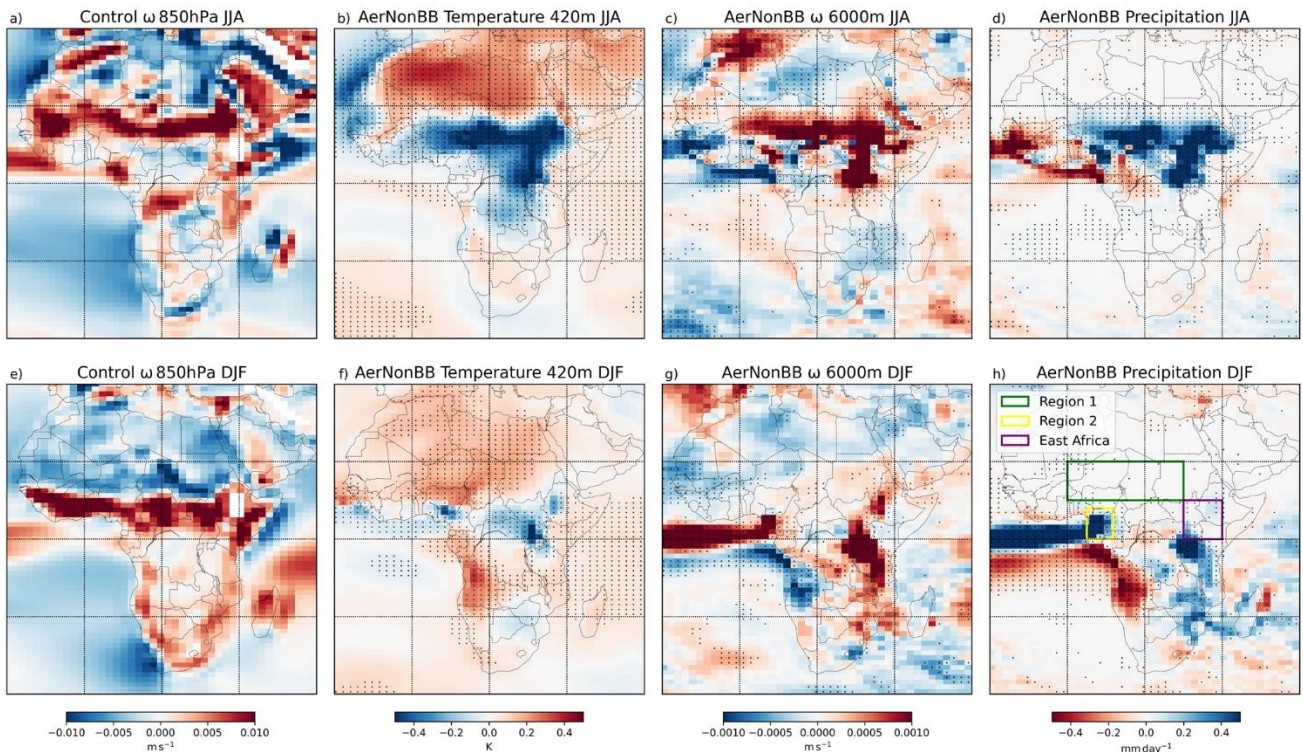

Figure 9: Background 850hPa vertical velocity in the control (left), and the AerNonBB-induced 2070-2100 changes temperature at 420m (2nd column), vertical velocity at 6000m (3rd column), and surface precipitation (right column), over Africa for JJA (top) and DJF (bottom). Stippling indicates significant changes when compared to the inter-member deviation.

The local precipitation response, while complex, is then intelligible when considering the patterns of both the emissions and the background circulation pattern. Similar features can be seen elsewhere, such as over East Africa, where emissions,

convection, surface cooling, enhanced mid-level convection, and rainfall are co-incident. Further examples of this mechanism are provided in the supplement for AerBB and the idealised perturbation experiments.

**4 Discussion and Conclusions**

This study analysed the effect on local and global climates of Africa pursuing SSP370 aerosol, reactive gases, and $CO_2$ trajectories rather than the strong mitigation SSP119 scenario. The responses are likely predominantly due to the substantial differences in aerosols, since aerosols have a larger forcing and more important role in precipitation than reactive gases. SSP370 comprises higher fossil fuel and biofuel use, with associated higher $CO_2$ emissions and non-biomass burning aerosols, with BB aerosol emissions conversely lower than in SSP119. The state-of-the-art CMIP6 Earth System Model UKESM1 was used in this study, with its two-moment aerosol scheme and interactive chemistry scheme simulating the response to changes in sulfate, organic carbon and black carbon aerosols, and reactive gases. Analysis of the transient scenario experiments was aided by several much larger idealised aerosol emissions perturbation experiments over Africa and the Tropics. These idealised experiments demonstrated large-scale responses consistent with the transient scenario-based experiments, including the net warming from carbonaceous aerosol and the broad dynamical responses of the ITCZ and precipitation effects.

While $SO_2$ and carbonaceous aerosol were changed simultaneously, the overall difference in $SO_2$ emissions between the experiments was small due to spatial cancellations; some local responses were driven by local $SO_2$ changes, but global effects and most prominent regional changes were driven by the carbonaceous aerosol response. UKESM1 features a strong BC forcing, causing the BC aerosol to be the dominant driver of the response. Increasing nonBB aerosol emissions from Africa therefore warmed the climate significantly more than in the control (by 0.07 K), with this difference enhanced to 0.22 K when including the additional African $CO_2$ emissions of SSP370 compared to SSP119. These global temperature changes are substantial when considering the disparate impacts between 1.5 K and 2 K of global warming (IPCC 2018).

The general warming was outweighed locally by cooling, due to the reduced surface incident radiation from increased aerosol. This local cooling persisted even when including the increased $CO_2$ concentrations. The pattern of this cooling did not match the emissions change, particularly in the northeast of the cooling region in JJA, as it depended also on the background circulation. Areas with both higher emissions and strong convection, particularly near the ITCZ over Africa, saw strong aerosol lofting and therefore a larger radiative effect, with consequent large local cooling. This lofted aerosol drove strong BC absorption, with a warming peak around 3-4km and enhanced instability above this, resulting in increased precipitation. The complex responses of temperature and precipitation, including the opposing local and remote temperature impacts, indicate the importance of investigating the regional response to specific emissions trajectories in addition to the global mean change.

A single model, UKESM1, was used in this project. While UKESM1 accurately simulates many aspects of the global climate, its representation of some aspects of the climate system relevant to this study is different to that of other models. In particular, it simulates a strong present-day BC forcing, the largest of eight AerChemMIP models at $0.37 \pm 0.03$ W/m2 (O'Connor et al., 2021; Thornhill et al., 2021); the multimodel mean result was $0.15 \pm 0.17$ W/m2. Its OC forcing is similar to other models; the

dominance of BC over OC in UKESM1 may therefore be at odds with other models. Of the six AerChemMIP models which calculated both OC and BC ERFs, UKESM1 is one of only two in which the magnitude of BC forcing outweighs that of OC. Thus the warming found here under higher African carbonaceous aerosol emissions is likely unrepresentative of most other models; similar experiments changing carbonaceous aerosols using other CMIP6 models would therefore be useful to investigate the model uncertainty in this response. However, UKESM1 and its earlier versions feature detailed and realistic

aerosol schemes, with the earlier version HadGEM3 demonstrating better observational BB properties than most models (Brown et al., 2021), and the aerosol scheme has undergone careful improvement and evaluation against observations (Johnson et al., 2016; J. Mulcahy et al., 2020; J. P. Mulcahy et al., 2018; O'Connor et al., 2021).

    The vertical profile of Black Carbon has been found to be biased compared to observations, with too much aerosol at high altitudes and not enough lower down, in CMIP6 models including UKESM1; when accounting for more realistic vertical

profiles, the BC forcing is reduced significantly (Allen et al., 2019). A more observationally-constrained simulation of these scenarios would therefore likely reduce the warming effect. Potentially, the mechanism analysed here through which precipitation is enhanced by the increased carbonaceous aerosol would be weaker under more accurate vertical profiles, though the extent of this is less certain without further experiments. Biases in underlying circulation patterns, such as the double-ITCZ bias (Tian & Dong, 2020), would also impact the specific pattern of precipitation increase found here.

The scenarios applied here changed subsets of emissions, whereas in reality specific anthropogenic activities and natural processes cause emissions of broad ranges of species. Using fossil fuel-linked aerosols from one scenario with $CO_2$ emissions of another neglects this link. In addition, the SSP scenarios include interactions between continents, contrary to the experiments here allowing the scenario to vary spatially. The scenarios created here are not designed to be specific plausible future pathways, but are instead generated to isolate the effects of a single continent varying in its trajectory of particular emissions

subsets. The SSP119 and SSP370 scenarios feature little difference in $SO_2$ emissions over Africa, due to the spatial cancellation visible in Figure 1. These scenarios were created by different Integrated Assessment Models, and this cancellation may not be a robust response. If these experiments were applied over other regions, or for other pairs of scenarios, sulfate aerosol may play a substantially larger role, with potentially larger local cooling and less remote warming.

    Emissions were changed only over a single continent, Africa, in the scenario experiments, with emissions elsewhere fixed at

SSP119 levels in all simulations. The experiments represent several possible future trajectories in emissions subsets, albeit following two very different emissions pathways spanning the full range of aerosol emissions in the SSP scenarios (Gidden et al., 2019). Only a subset of emitted species was changed each time – either some or all aerosols and reactive gases, with and without $CO_2$. These relatively small perturbations to total emissions nonetheless generated substantial impacts on the climate, both locally and at the global level. The large effect of the different scenarios reflects the far wider range in aerosol emissions

trajectories in the SSP scenarios used here compared to the previous RCP scenarios. This stresses the importance of aerosols for the future evolution of global and regional climate, and simultaneously highlights that further attention should be paid to the impact of future African sulfate and carbonaceous aerosols in particular, since they can lead to substantial modulations of the predicted patterns of future temperature and rainfall for different world regions.

This study highlights the importance of future African aerosol emissions for local and global climates, showing that the range

of plausible futures over a single continent can drive significant shifts in global temperatures, and substantial impacts in local and remote temperatures and precipitation. Further work should be undertaken to better understand the plausible future variation in aerosol-climate impacts, exploring a range of scenarios, regions, species, and models.

**Data availability**

Reasonable requests for model output and the data used for figures in this manuscript are available from the corresponding author.

**Author contributions**

AV and CDW conceived designed the experiments. CDW carried out the analysis with assistance from MK and led the

470 preparation of the manuscript. All authors provided discussion of the analysis and reviewed the manuscript.

**Competing interests**

The authors declare that they have no conflict of interest.

**Acknowledgements**

This work was supported by the Natural Environment Research Council (grant number NE/L002515/1). AV and MK acknowledge the Leverhulme Trust for providing funding through the Leverhulme Centre for Wildfires, Environment and Society. Simulations with UKESM1 were performed using the Monsoon2 system, a collaborative facility supplied under the Joint Weather and Climate Research Programme, which is a strategic partnership between the Met Office and the Natural

Environment Research Council. The authors thank Malte Meinshausen for kindly providing $CO_2$ concentrations with MAGGIC6.

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
