# Peer review of "Local and remote climate impacts of future African aerosol emissions"

_EGUsphere, 2022_

## Author Comment (AC1)

Many thanks to both reviewers for their useful comments and feedback; we have added our responses in blue text below each point.

**Reviewer #1**

**General comments**

The authors of this manuscript have conducted a study of the impacts of different future African aerosol and $CO_2$ emission pathways on global and local climate. This is done through simulations with one Earth system model (ESM), UKESM1, using several ensemble members for each experiment. They find that increased future non-biomass burning carbonaceous aerosols in Africa lead to warming on the northern hemisphere, northwards shift of the Inter-Tropical Convergence Zone and precipitation changes in the vicinity of the emission changes, compared to their control experiment. The authors present a detailed analysis of the mechanism behind the local precipitation response in the emission region.

Climate impacts of future aerosol emissions over Africa, the topic presented in this study, is relevant for the climate research community and readers of Atmospheric Chemistry and Physics, and, as the authors claim, not yet well studied. The method the authors use is well described and appropriate for the study.

The manuscript is in general well structured but the writing and presentation need a major revision. The whole manuscript needs to be reworked for clarity an preciseness (including abstract, figure captions and supplement material) but in particular the result section, and the discussion and conclusion section for clarity and conciseness, please also see specific comments below.

The authors have presented an interesting and well conducted study, but are asked to address the following questions/comments:

I miss a motivation for why the authors choose to focus on the results from the scenario simulations. At least one sentence describing the motivation behind this choice should be added.

This is an important point; we have extended the final paragraph of the introduction to address this. Specifically we have noted that *"These experiments are focused on due to the societal importance of realistic future emissions scenarios".*

Figure 1 where the authors present the aerosol emission in the different experiments is very confusing, and the authors need to rethink how the emissions are presented. Please also see specific comment below.

We have redone Figure 1 to make it much clearer, by labelling each non-control curve with the experiment(s) that it applies to.

There is no description of the radiative forcing or effects on radiation generated by the different aerosol (or $CO_2$) emissions in the manuscript, apart from the short-wave radiation absorption profile in two limited regions. The authors describe competing effects of reflecting (cooling) organic carbon and absorbing (warming) black carbon, but never present a quantification of these effects. Moreover, the authors describe an ESM with a climate sensitivity larger than that of other ESMs participating in CMIP6. Therefore, it is relevant to present the radiative forcing or changes in atmospheric radiation in the simulations, and perhaps the multi model mean sensitivity of the CMIP6 models when discussing the hight climate sensitivity of the model used in this study.

We have added further context on the high ECS of UKESM1, by adding the CMIP6 multi model mean and the very likely AR6 range. The effective radiative forcings of the scenario experiments are unavailable, as we did not run the fixed-SST experiments necessary to do this, but we did perform these for the idealised experiments. We have therefore added the global mean TOA, surface, and atmospheric forcings for the 10x Tropical SO2 and OC+BC experiments in the supplement. To provide further context on the scenario experiment impacts on radiation, we have added a paragraph describing and discussing the effects of the AerNonBB and AerAllCO2 experiments on the clear-sky radiative flux components into the main text.

The authors cite literature that describe the influence of aerosols on global atmospheric circulation and in particular the Asian monsoon, but, despite being a study of African aerosol emissions and associated local and global effects on the climate, there is no discussion or citations of studies regarding the African monsoon and previous studies of climate impacts of African aerosols on regional or global scale. The authors need to present more background regarding African aerosols and climate impacts and African monsoon and cite relevant previous studies on this topic.

We decided to focus on regions of rainfall away from the West African Monsoon in this study due to the clearer impacts and mechanism, and the substantial intermodel variation in the monsoon and its future. We agree that the west african monsoon should still be discussed and analysed briefly for context in the paper. We have therefore added an additional paragraph in the introduction describing past research on the impact of aerosols on this monsoon system, and also some analysis in the results on the effects of the scenario experiments.

The high sensitivity of the particular ESM used in this study warrants a more elaborate discussion of the impact on the results in relation to the choice of using future scenarios (SSPs) and not idealised emission perturbations, and the simulated

aerosol burden and forcing (which is not shown in the manuscript). The authors should present more numbers (e.g. radiative forcing, sensitivity) to illustrate how this model responds to changed emissions compared to other models participating in CMIP6.

In addressing an earlier point we have added several parts which also address this area: results of the global mean forcing and fluxes for the idealised and scenario experiments respectively (as forcings aren't available for the scenario experiments), and further context on CMIP6 and AR6 ECS values. We have also added the AerChemMIP multimodel mean BC value to the discussion to provide further context to UKESM1's value in that discussion point.

The authors need to work on making the language more precise and at times more concise. The disposition of the Discussion and Conclusions section is not optimal, where discussion and conclusions are mixed and presented in an order that is not always logical.

We have reorganised parts of the Discussion and Conclusion section, and added some further context from other studies and other clarifications to better describe this section.

**Specific comments**

Given the particular high sensitivity of the single model used in this study, it might be appropriate to change the title to "Local and remote climate impacts of future African aerosol emissions in UKESM1", to highlight the fact that this study portrays the result of this particular model, which is not necessarily representative for what a similar study using a different model might show.

We prefer on balance to keep the title in its current, more concise form. While specifying the model in the title may be useful context in some cases, many readers would likely have to move to the abstract to know the context of the model (i.e. that it is a CMIP6 generation model). Prior single-model papers typically don't note this aspect of the study in the title. We think the intermodel context within the paper sufficiently situates the nature of the study within the broader literature.

The figure/table captions are inconsistent throughout the manuscript. In some of the figures the indication of whether a result is statistically significant or not is described in the figure caption, sometimes it is described in the text. Sometimes a capital letter is used at the start of the figure captions sometimes it is not. The authors should go thorough all the captions to make sure they describe the figures/table in a consistent manner.

We have moved the definitions of significance in each figure and table to the caption for consistency.

Line 12: Please change "investigating" to "to investigate".

We have made this change.

Line 14: "sees" should be rephrased

We have made this change.

Line 14: What is meant by "direct anthropogenic"? And are not biomass-burning emissions anthropogenic?

We have clarified this and made it consistent by referring to them as non-biomass burning emissions.

Line 15: The reduced short wave surface radiation is never shown in the manuscript.

We agree that the manuscript would benefit here and elsewhere from displaying the SW surface radiation response in the scenario emissions, and have therefore included this as the new Figure 3, before the spatial temperature information.

Line 16: What do the authors mean here? That absorption from black carbon takes place away from the emission region?

The black carbon absorption generates a positive TOA forcing, which outweighs the organic carbon scattering effect (as shown in the newly added idealised ERFs in the beginning of the supplementary material). This drives an overall temperature increase, widespread as the atmosphere redistributes the additional heat. The local effect is that the reduced surface radiation due to the increased aerosol above the main emissions areas is large enough to outweigh this general warming effect. We agree that the original wording of this sentence was convoluted and have reordered it to note the more general effect first, and this local response second.

Line 16: Can the authors provide a quantification of the global warming here?

This is a very useful addition to the abstract; we have added numbers from Table 1 (note this is now Table 2 as we have added a table to describe the scenario experiments as suggested by the other reviewer) to note the significant global warming relative to the control under AerNonBB and AerAllCO2.

Line 18: The authors should be more specific regarding how the ITCZ changes, i.e. northward shift due to a warmer northern hemisphere compared to the control experiment.

We have changed the abstract to make this more specific.

Line 27-28: Please provide a context for the scenarios. For example that they are used for simulating future climate impacts with general circulation models or similar.

We have added a sentence to note this in this paragraph.

Line 29-31: This sentence is very difficult to read, please rephrase.

We have split this into two sentences and rephrased them to be clearer.

Line 51-52: I am note sure what the authors mean by this, they need to specify what emission trajectory differs in contrast to what. There are definitely published studies where specific regions follow an aerosol emission trajectory different from the one specified for the CMIP5.

We have clarified this to refer directly to SSP emissions in CMIP6 models.

Line 58: This sounds more like the hypothesis you are set out to investigate, please change to something like "These emissions could therefore have a substantial effect on local and remote climates."

We agree that the suggested wording is better and have changed this sentence.

Line 58-60: This sentence is not very clear and could be rephrased to be easier to read.

We have rephrased this sentence to improve it.

Line 79: Please be more specific when describing the model's performance that stating the it represents the climate "well" (and "reasonable"). A quantification of differences, e.g. sensitivity, forcing etc.

We have further situated UKESM1 in terms of CMIP6 and other climate sensitivities by adding the CMIP6 multimodel mean and AR6 range. We have also included the UKESM1 and AerChemMIP multimodel aerosol forcings for another quantitative comparison. We have also added additional context on the historical temperature timeseries.

Line 80-84: Repetition of "its". Please rephrase.

We have rephrased this paragraph.

Line 103-104: What is meant in this sentence, please clarify. Is there no biomass emissions of $SO_2$? What impact does this have on the result regarding

absorption/scattering dominating the radiative effect of in the different experiments?

UKESM1 doesn't represent biomass emissions of SO2, and we have clarified this sentence to underline this. The UKESM1 team justified this as they had tested the model without SO2 BB, so including it would necessitate re-testing, and since present-day SO2 BB emissions are around 2% of anthropogenic emissions this effect would be small anyway. The nonBB experiment would naturally be unaffected by including SO2 BB, but assuming SO2 BB emissions are lower along with carbonaceous BB in SSP370, the BB experiments would feature less cooling than actually found due to a slightly lower SO2 forcing. This would also add a slight warming to the AerAll and AerAllCO2 experiments. But since SO2 BB emissions are very low, this change would be slight.

Line 105: Should it not be "'AerAll' refers to the experiment where all aerosols and reactive gases follow the SSP370 scenario over Africa" or similar? Please go through this paragraph to make sure that the experiments are described properly, and that the references to the different experiments are consistent with the result sections, i.e. are you referring to the aerosols or the experiment? As the text reads now, there is no consistency regarding this.

We have put the scenario experiment definitions in a Table as suggested by the other reviewer, and have reworded parts of this paragraph to additionally make the text clearer.

Line 116: I doubt that climate equilibrium was reached, rather that the ocean mixed layer has adjusted to the imposed forcing.

We have reworded this sentence to remove the reference to equilibrium, as it is correct that the full climate system will not have fully equilibriated by this point.

Line 132: What is meant by "larger three experiments"? Should it be the experiments with large emission increases?

We agree that the suggested wording is better and have used this.

Line 132-136: Do the authors have an estimate of how much this might influence the results?

We did not have the computing resources to verify that the effect of these changes is small, but we would expect the effect to be secondary as the main drivers of CH4 and vegetation effects (i.e. CH4 emissions and the CO2 physiological effect) are unchanged in these idealised experiments.

Line 137-153: This paragraph and Figure 1 need some reworking/clarifications, and better connection between the emission descriptions in the text and what is shown in the figure. Figure 1 would benefit from labeling of each panel and more specific references in the text. It is difficult to understand which panel describes which experiment and how to interpret the figure.

The black solid lines in the line plots describe the emissions in the control/SSP119 of different aerosols and $CO_2$. Likewise, black dashed lines describes African emissions in control/SSP119. My interpretation is that the emissions in "AerAll" are described by "OC+BC total" and "SO2", but this in not obvious. Moreover, when it comes to the "AerBB" and "AerNonBB" experiments, it is no longer clear to me how I should read the total aerosol emissions in these experiments from the figure. I understand that "OC+BC BB" shows the change in biomass burning aerosols, but how do the total aerosol emissions change, including $SO_2$? I also do not see what the map plot "All 2xBB" should represent in this context.

The authors should rethink how the emissions are presented in Figure 1. What does "perturbation" represent in the legend in each panel?

We have substantially reworked Figure 1 to better display the emissions scenarios. This includes adding panel labels, as we have for all figures, to aid in referring to the figures in the text.

Line 159: Is Figure 1 showing actual $SO_2$ emissions or perturbation/control ratio?

There was a mistake in the caption suggesting it is the ratio, but the graph shows the total emissions – thanks for highlighting this issue; we have edited the caption to reflect the graph now.

Line 168-170: This sentence seems out of place here. What is the point of the table? What does the table add to the story? It would be better to first focus on Figure 2 and then discuss the table. Where does the pi-control data come from?

Table 1 quantifies the pre-industrial to 2070-2100 global and selected regional temperature changes under the different scenarios, and analyses the significance on each of these scales. It serves to show the effect on overall global and regional warming under each scenario, and the effect of following the different Africa aerosol and CO2 scenarios on these outcomes. For example it estimates that 2.27K of overall warming would occur under SSP119 in UKESM1, but that this would be 2.5K if Africa follows SSP370. We felt that the global data from Table 1 should be referred to in the discussion of the global mean temperatures in Figure 2, as it allows the reader to quantify the difference between the curves and to understand the level of significance between them. We have reworded this section to clarify the comparisons, as suggested in the next point, to clarify the text and the motivation. A

pre-existing PiControl member was used for the pre-industrial temperatures; this information has been added to the manuscript.

Line 172: What is cooling in comparison to what? The whole result section suffers from this lack of clarity and needs to be carefully reworked so that is clear in everywhere what is being compared. The authors also have a tendency to describe the figures rather than letting the figure support the result they are describing.

We have reworded these paragraphs to more clearly state the comparisons, and to note the overall global warming levels in several experiments.

Line 181: Higher warming in comparison to what? Is the difference of 0.16K between the $CO_2$ experiment and the control/SSP119 scenario? Why is Table 1 referenced here which shows the temperature differences between the experiments and a pre-industrial state?

We have replaced this value with the actual warming level in the experiment to match the table, and have clarified the comparisons.

Line 184: Warming compared to what?

We have now clarified this.

Line 190: Please change "plot" to "panel", and check the other figure captions as well.

We have made this change and changed all other references to this wording. As suggested by the other reviewer we have added panel labels to each figure and referred to individual panels in the text.

Line 197: I think it would be more appropriate to explain the red/blue numbers here than in the text.

We agree and have incorporated this change.

Line 200: It has not been shown here where the aerosol-burden change is largest or radiative effect of the changed aerosol emissions.

We have added a plot of changes in SW surface radiation as suggested above, and hope this, along with the global mean radiation changes also introduced, gives sufficient context to this section.

Line 210: To me it looks like the warming is more prominent in south western Africa. I can't see a statistical significant warming in South eastern Africa neither in DJF nor in JJA. Could the authors clarify what is meant here?

This was an error in the initial manuscript, which should have read "southwestern". Thank you for finding this; we have now corrected it.

Line 214-219: This paragraph seems to belong together with the discussion about the global mean temperature response before line 198.

We feel that these regionally aggregated results are better understood with reference to the spatial patterns, as the reader can more easily understand the different response over the emissions region compared to more remotely. This is especially the case in NTropAf, which was defined from the spatial maps to quantify the strong cooling over this region – this would we feel be less well motivated if introduced before the temperature response maps.

Line 216-219: This sentence is confusing. The temperature change over Africa is compared to that of the global mean relative to the pre-industrial(?) simulation, and the contrasted to temperature changes relative to the control. Please rephrase.

We have rephrased this sentence.

Line 220-221: Cooling compared to what?

We have clarified this point.

Line 228: Which season?

We have removed these words as the season DJF is noted further in the sentence.

Line 230: What is meant by "north of this"?

We have clarified by changing to "north of the DJF position".

Line 239: What is referred to by "this"? Please be more specific. Please remove "850hPa map" and specify which figure/panel is being discussed.

We have changed to "this southward shift" and added panel references.

Line 240: Please rephrase "hints of northward shifts". Which figure is discussed here?

We have rephrased and added a panel reference.

Line 254: "In the AerAll and AerNonBB experiments"

We have changed this.

Line 256-257: What do the authors mean here? Please clarify. How does the position of the ITCZ influence the strength of the forcing asymmetry?

The asymmetry relevant for shifting the ITCZ is defined across the ITCZ, rather than the equator itself. Therefore the position of the ITCZ relative to the aerosol forcing plays a role, particularly here where the aerosol emissions are close to the ITCZ. We have changed the misleading use of "interhemispheric" to improve this.

Line 258: Discussed presently?

Changed to "discussed shortly".

Line 258-260: This is also true for AerAll experiment.

That is correct and we have now specified this; the prior version implied that the changes are found in AerNonBB and AerAll as nonBB aerosol is increased in both, but it reads clearer now that we have specified these two experiments.

Line 260-261: This sentence does not make sense to me. "ITCZ impacts can produce substantial changes in locations remote from their emissions"? Please rephrase.

We have rephrased to "through their impacts on the ITCZ location, aerosol emissions can produce…".

Line 273: How can this be three pairs?

We have changed this wording to "combinations" instead, in this case and all others.

Line 276-278: This sentence seems redundant.

We feel that this paragraph is needed to introduce both figures 6 and 7 to the reader to help them to follow the next section, which relies on aspects of both. This sentence describes figure 7 as the prior sentence does for figure 6, so we feel that it should remain as part of the paragraph (note these figure numbers have now changed from 6 -> 8 and 7 -> 9 as figures have been added through the review process).

Line 280: Please add labels for each panel in Figure 7 and specify which panel is discussed.

We have now added panel labels to all figures to aid the reading of the paper.

Line 284: What role does the seasonality of precipitation have in the aerosol burden in these areas?

This is an interesting question, but these effects would be hard to disentangle from the effect of seasonal cycle in emissions. This would require further experiments with no seasonal emissions cycle. This would also be entangled with the effect of the changed aerosols on precipitation.

Line 286: What is meant by "since the amount of aerosol present to block radiation decreases with altitude"? The near-surface cooling is most likely a consequence of changed surface energy fluxes.

This is a good point, and we have removed this bracket to account for this.

Line 289: Which panel if Figure 7?

We have added panel references to the figures and references in the text.

Line 298: Please add a reference to Figure 7 and which panel specifically.

We have added this reference.

Line 298-299: Where is this result shown, and what do the authors mean? That aerosols transported over the Atlantic lead to enhanced precipitation locally?

We are noting that the effect seen in Region 2 in DJF also occurs downwind into the Atlantic, as a result of the transport of the aerosol over the ocean – now Figure 9h.

Line 333: To my it sound strange to write that "This study analyses". Likewise, that "scenarios sees changes".

We have rephrased these wordings to improve them.

Line 334: Please remove "bulk" and rephrase.

We have done this.

Line 337: Should it be "two-moment scheme" here?

Thank you for finding this error; we have changed it to the correct wording.

Line 360: Perhaps "locally in the emission regions"? But the pattern did not match the emission strength? Please be more specific, where is the cooling and in what sense does it not match the emissions?

In the northeast of the cooling region in JJA, there are regions of strong cooling which do not lie within the strongest emissions regions. We decided to use annual emissions maps to display the emissions to avoid showing too many plots, so a

direct seasonal comparison isn't possible within the paper. We have added to this part to clarify this region and season.

Line 372: ERF has not been introduced previously in the text.

We have now introduced ERF in the introduction, and added the acronym definition there.

Line 376: What is meant by accurate in this context? Detailed or sophisticated perhaps?

We have changed to "detailed and realistic".

Figure 3: The global maps are very small and it is difficult to see which temperature changes are statistically significant. The panel showing the temperature change due to changed $CO_2$ emissions from Africa is never discussed in the text and should perhaps be removed.

We have improved the stippling in this plot (note this is now figure 4). We feel the CO2 panel should be retained for completeness and have added a reference to this panel in the text, noting its relative uniformity compared to the aerosol response.

Figure 4: These map plot are even smaller then those in Figure 3 and are even harder to read. Perhaps the authors should split the map plots and pressure/latitude plots into two different figures.

We have split this figure into separate maps and pressure/latitude plots as suggested; these are now Figures 5 and 6.

Figure 5: Please see comment regarding Figures 3 and 4.

We have improved this figure, but we feel that it is useful to have the zonal mean plots next to the maps as they share a common axis, aiding the reader in visualising the overall effect spatially and latitudinally.

Figure 6: The description of how the profiles of stability and vertical velocity change relative to the control experiment and whether they are significant or not is difficult to follow. Is there not a better way of presenting this, perhaps by showing the profiles from the control experiment? Omega ($\omega$) is usually used to denote pressure velocity, and is never introduced in the text, neither is $\theta$.

We have removed the different shades from the theta plot as the domain shown is always unstable. We have added the control lines to the vertical velocity plot. We have also clarified the symbols used in the caption.

**Technical corrections**

Thank you for highlighting these; we have corrected them.

Line 43: Remove parentheses around reference at the beginning of this sentence.

Line 53: The acronym CMIP6 has not been introduced.

Line 64/66: Repetition of "inform/ed"

Line 65: Please change "bulk" to something more elegant.

Line 85: Insert space before reference.

Line 100-102: and throughout the text: SO2 to $SO_2$ , NH3 to $NH_3$, C2H6 to $C_2H_6$, C3H8 to $C_3H_8$, C3H6O to $C_3H_6O$, C2H4O to $C_2H_4O$

Line168: Are these temperature differences from the scenario experiments?

Line 171: "Run" is jargon.

Line 177: Space between forcing (O'Connor

Line 227: remove "upwards"

Line 228: Change circulation to vertical velocity

Line 274: bottom right panel of Figure 7

**Reviewer #2**

General comments:

The authors use a single coupled climate model to estimate climate impacts from African aerosol perturbations. This is certainly an important topic that has not been given much attention, as the manuscript points out. Though the manuscript has

some interesting results, I found several of the figures and conclusions to be fairly confusing at this stage. I would like to see the authors clarify these points in the specific comments before recommending publication.

Though this is a global modeling study, the results are mostly focused over Africa. Yet there are no African authors. I note that the authors here tease a future paper on air quality impacts. I highly suggest they begin collaborating with African scholars and experts to avoid perpetuating colonialism in science.

We agree wholeheartedly on the importance of collaborating with researchers in the regions of study, and that the neglect of this historically is reflected in the limited amount of similar prior studies focusing on Africa. We regret that this project is now unable to do so, as it was limited in scope as part of a PhD project which has now completed.

Specific comments:

Line 39-40: provide some citations for the statements about aerosols impact on precipitation. Also, "scattering aerosols tend to decrease precipitation…" I suppose a more accurate statement would be "Increases in scattering aerosols tend to decrease precipitation". Likewise for the black carbon statement in the following sentences.

We agree the suggested wording is better and have used this in these cases. We now have three references in this paragraph supporting the statements about aerosol-precipitation interactions.

Line 57-58. I'm not following the how the first sentence "The tropics cover half…" implies the second sentence, that they will have substantial effect on remote climates.

We have changed the second sentence to reflect this.

Line 60: What's a direct human health impact?

We were referring to air pollution impacts rather than through the climate impacts of the aerosols; we acknowledge this was not clear and have rephrased to address this.

Line 79-85: There needs to be a little more effort dedicated on this model evaluation. I can understand not wanting to perform a model evaluation for this paper since it has already been done in other papers, but we at least need to know some quantitative values rather than just statements like representation is "reasonable".

We have added further context with CMIP6 and AR6 ECS values, and further AerChemMIP aerosol forcing values for comparison.

Line 101: suggest using the more common "Formaldehyde" for HCHO

We have made this change.

Line 100-127: Suggest putting all of this in a table, or otherwise nicely organizing it. Right now it is confusing.

We have followed this useful suggestion to put the scenario experiments in a table (now Table 1), and have reworded this section too.

Line 112: Need to explain how the ensemble members are different

This is important information and we have now added this.

Figure 1 and accompanying text: It is very confusing. I think you need to label things as either "AerAll" or "AerBB" rather than just "perturbation". What is "All 2xBB" row showing? Hard to make much sense out of this figure.

We have changed Figure 1 substantially to better communicate the emissions used under each scenario experiment.

Line 166: and reducing variability by averaging over ensemble members, right?

Yes; we have added to this sentence to note this.

Line 180-185 and Figure 2. The figure here makes more sense than Figure 1. However, I'm having trouble following the description. AerNonBB means that fossil fuel and biofuel aerosol and reactive gas emissions follow SSP3-7.0 in Africa, but everything else follows SSP1-1.9. So there's more surface warming with AerNonBB? Is this because of Black Carbon? According to Table 1, the local effect over Africa is a cooling, so where is that warming coming from relative to the control?

Figure 2 shows the global temperature responses, which are also shown in Table 1 (note this is now Table 2). The global response to increased nonBB aerosol in AerNonBB is a higher temperature than in the control, due to BC absorption. This is what is shown in the Figure 2. Africa instead exhibits a cooling due to the reduced incident radiation, as indicated in Table 1 and Figure 3. We have substantially reworded this section to make it much clearer which experiments and spatial regions are being compared at each point.

Figure 2: I was looking for the results of AerAll, is it on this plot at all? Perhaps it is and the bright yellow doesn't show up well.

AerAll is in yellow, which was not clear enough on the original plot. We have changed the colours to improve this, and also used a colourblind-friendly palette. AerAll is similar in its response to AerNonBB due to the weak effect of the BB aerosol difference between them.

Line 200: Ok, here is some explanation regarding the previous results in Fig 2 and Table 1. Suggest moving this up. Still not seeing why is there so much warming in remote areas in AerNonBB and AerAll?

We feel that the regional information should be introduced after the global information as it is more complex. There could be some confusion caused by introducing Table 1 with Figure 2, to discuss the global response, before Figure 3, as Table 1 also includes regional information. But the alternative would be to split into two tables for global and regional effects, and we feel this would clutter the paper and de-emphasise the result that experiments can see significantly different temperatures from the control in specific areas while not having a significant difference globally (e.g. AerAll). We hope instead that the rewording we have done of this section provides sufficient context to readers on the various comparisons. The remote warming in AerNonBB and AerAll is due to the redistribution of the additional heat from BC absorption.

Fig 3 and 4 (and all figures really): recommend having labeled panels a,b,c,d,e, ... etc. Will clear things up when you can refer to a specific panel

We have added panel labels to all figures and referred to the specific panels in the text.

Line 234: But we haven't actually seen the radiative forcing patterns yet in this paper, have we? And shouldn't the forcings be pretty close to the emissions changes, which are all in the tropics?

We aren't able to include radiative forcing plots for these experiments, as we didn't have the resources to simulate fixed-SST experiments to calculate these forcings. We feel that adding additional TOA flux plots would not be justified, but the surface SW forcing plot we have added shows the annual hemispheric asymmetry. It is true that the forcings are close to the tropics and hence the ITCZ, but an asymmetry still occurs in some seasons, which drives shifts in the ITCZ location.

Figure 4: is there supposed to be stippling to indicate significance on the 850 hPa omega plots? I see it on the zonal plots.

There is, but they were not very visible. We have split this plot into 2 figures (now figures 5 and 6), following the other reviewer's comments, and improved the stippling so this is more visible.

Line 256: I'm not following how the more norther position of the ITCZ in JJA prevents the perturbations from inducing a strong interhemispheric forcing asymettry.

The original wording was incorrect, as it used "interhemispheric". The determining factor for an ITCZ shift is a forcing asymmetry across the ITCZ, not the hemispheres separated by the equator. This ITCZ asymmetry is naturally affected by the ITCZ location relative to the emissions & hence forcing; we have reworded this sentence to address this.

Line 274: is there something unique about these regions that warrants further focus, or is it just because they were "hotspots"

These are hotspots of precipitation response in societally important regions, especially the highly populated Region 2. These didn't align with the main emissions areas, warranting further investigation of the mechanism of these responses.

Line 334: Unless I'm missing something, the reactive gases aren't really ever mentioned much in the results. So what is this statement based on?

We did not have the resources to perform sensitivity tests with only reactive gases, but we wanted to include the full set of short-lived species to be consistent with the scenarios. Given the small forcing of NMVOCs compared to aerosols in AR6 and AerChemMIP, and the key role of aerosols in precipitation changes, we think it very likely that the main effects are due to the aerosol changes. We wanted to note this so as to not distract the reader with the reactive gases, but we do not have the evidence to confirm this, so we have changed this sentence to reflect that we expect the effect to be dominated by the aerosol response.